# Kinetic Studies of Antioxidant Properties of Ovothiol A

**DOI:** 10.3390/antiox10091470

**Published:** 2021-09-15

**Authors:** Nataliya A. Osik, Ekaterina A. Zelentsova, Yuri P. Tsentalovich

**Affiliations:** 1International Tomography Center SB RAS, Institutskaya 3a, 630090 Novosibirsk, Russia; n.osik@tomo.nsc.ru (N.A.O.); zelentsova@tomo.nsc.ru (E.A.Z.); 2Physical Department, Novosibirsk State University, Pirogova 2, 630090 Novosibirsk, Russia

**Keywords:** ovothiol, antioxidant, glutathione, triplet quenching, rate constant

## Abstract

Ovothiol A (OSH) is one of the strongest natural antioxidants. So far, its presence was found in tissues of marine invertebrates, algae and fish. Due to very low pKa value of the SH group, under physiological conditions, this compound is almost entirely present in chemically active thiolate form and reacts with ROS and radicals significantly faster than other natural thiols. In biological systems, OSH acts in tandem with glutathione GSH, with OSH neutralizing oxidants and GSH maintaining ovothiol in the reduced state. In the present work, we report the rate constants of OSH oxidation by H_2_O_2_ and of reduction of oxidized ovothiol OSSO by GSH and we estimate the Arrhenius parameters for these rate constants. The absorption spectra of reaction intermediates, adduct OSSG and sulfenic acid OSOH, were obtained. We also found that OSH effectively quenches the triplet state of kynurenic acid with an almost diffusion-controlled rate constant. This finding indicates that OSH may serve as a good photoprotector to inhibit the deleterious effect of solar UV irradiation; this assumption explains the high concentrations of OSH in the fish lens. The unique antioxidant and photoprotecting properties of OSH open promising perspectives for its use in the treatment of human diseases.

## 1. Introduction

Ovothiol A (1-methyl-4-thiol-L-histidine, OSH; Figure 1) is a naturally occurring antioxidant with unique properties. Firstly, despite its low molecular weight (only 24 atoms), OSH has four distinct basic sites, which correspond to 32 site-specific basicities [1,2]. Secondly, the SH group of OSH has very low pKa value (pKa ≈ 1.0–1.4) [1,2,3,4], in comparison with other natural thiols, such as glutathione GSH (pKa = 8.7), cysteine (pKa = 8.4) and coenzyme A (pKa = 9.8). Thiols act as good electron donors only in anionic thiolate form and, under physiological conditions, only a small fraction of GSH is deprotonated, while OSH is present almost entirely in the thiolate form. For that reason, OSH reacts with oxidants (Fremy’s salt, ferricytochrome c, H_2_O_2_ and tyrosyl radical [3,5,6,7,8,9]) much faster than GSH, yielding oxidized ovothiol OSSO. At the same time, the redox potential of OSH is more positive than that of GSH and the equilibrium in Equation (1) is shifted well to the right [4,8,10,11].
OSSO + GSH ⇄ GSSG + OSH.(1)

In biological systems, OSH and GSH act in tandem, with OSH neutralizing oxidants, such as reactive oxygen species (ROS) and free radicals, and GSH maintaining ovothiol in the reduced state [12,13]:(2)ROS + OSH → stable products + OSSO,
(3)OSSO + 2GSH → 2OSH + GSSG,
(4)$$\mathrm{GSSG}\;+\;\mathrm{NAD}(P)H\;\overset{\mathrm{glutathione}\;\mathrm{reductase}}{\to }\;2\mathrm{GSH}\;+\;\mathrm{NAD}(P).$$

OSH was first discovered in eggs of marine invertebrates [14,15,16,17,18]. It has been suggested [7,11,17,19] that the primary role of OSH is the protection of eggs from the deleterious effect of H_2_O_2_ produced during the respiratory burst of fertilization. Later on, OSH was found in other species, including marine algae [20,21], marine ragworm [22] and intracellular pathogens [23]. Biosynthesis of OSH is based on the coupling of histidine and cysteine molecules; the reaction is catalyzed by enzyme OvoA [12,24,25]. Evolutionary analysis of OvoA in metazoans [25] has shown that the gene coding OvoA was lost in bony fishes. Nevertheless, OSH was unexpectedly found in fish tissues [13,26,27]. Especially high concentrations of OSH (up to several mM) were observed in the fish lens. Most likely, a fish acquires OSH with food and accumulates OSH in the lens using OSH-specific transporter proteins. This finding indicates an important role of OSH in the protection of the fish lens—a tissue subjected to the irradiation by UV-visible light.

The deleterious effect of solar irradiation on biological tissues includes two major mechanisms, light-induced ROS generation and formation of highly reactive triplet species able to react with lens proteins. Since the lens tissue mostly consists of fiber cells without inner cellular apparatus, lens protection relies on metabolites. For example, the protection of the human lens from solar irradiation is provided by UV filters (kynurenine and its derivatives) absorbing light in the UV-A region, triplet-state quencher ascorbate, and glutathione being able to effectively reduce free radicals formed under irradiation [28,29,30,31]. The fish lens does not have UV filters (more precisely, no UV filters in the fish lens have been found so far) and the ascorbate level is rather low [26,27]. Thus, one can assume that high concentrations of OSH in the lens indicates not only its ability to reduce ROS, but also the ability to quench light-induced triplet states.

For a better understanding of chemical and biochemical processes occurring in the fish lens under oxidative stress or UV irradiation, one needs to know the mechanisms and the rate constants of both the thermal reactions of OSH oxidation and OSSO reduction and of the photochemical reactions involving OSH. This work is aimed at the determination of rate constants of elementary steps of antioxidant activity of OSH: OSH oxidation by H_2_O_2_, OSSO reduction by GSH and triplet-state quenching by OSH. To this end, we performed direct kinetic optical measurements and laser flash photolysis (LFP) experiments. The kinetic optical measurements were carried out with the temperature variation from 5 °C to 25 °C in order to estimate the Arrhenius parameters of the reactions under study and the pH variation from 6.8 to 7.4 for the evaluation of the influence of pH on the observed rate constants.

## 2. Materials and Methods

### 2.1. Materials

Kynurenic acid and reduced glutathione from Sigma-Aldrich (St. Louis, MO, USA), chloroform from Chimmed (Moscow, Russia), methanol from Merck (Darmstadt, Germany), D_2_O 99.9% from Armar Chemicals (Döttingen, Switzerland) and 10% aqueous hydrogen peroxide solution from Panreac (Vallès, Spain) were used as received. H_2_O was deionized using an Ultra Clear UV plus TM water system (SG water, Hamburg, Germany) to the quality of 17.8 MOhm. Chemicals for HPLC were purchased from Sigma-Aldrich (St.Louis, MO, USA) and Cryochrom (Petersburg, Russia). All samples were prepared in 20 mM phosphate buffered saline (PBS).

The disulfide form of ovothiol A was isolated from the lens tissue of pike-perch (*Sander lucioperca*), as described previously [13]. Briefly, the lens tissue was homogenated with a TissueRuptor II homogenizer (Qiagen, Venlo, The Netherlands) in cold (−20 °C) MeOH and then water and chloroform was added so that the volumetric ratio of water:chloroform:methanol was 2:2:1. The mixture was shaken in a shaker for 15 min and left at −20 °C for 30 min. Then, the mixture was centrifuged at 16,100× *g*, +4 °C for 30 min and the upper (MeOH–H_2_O) layer was collected. To provide only one form of ovothiol A in the extract, namely OSSO, hydrogen peroxide was added to the solution (10 μL of 10% aqueous solution per 1 mL of extract). The isolation of ovothiol A from the extract was performed by HPLC. The LC separation of the metabolomic fraction and collection of OSSO were performed on an UltiMate 3000RS chromatograph (Dionex, Germering, Germany) using a hydrophilic interaction liquid chromatography (HILIC) method on a TSKgel Amide-80 HR (Tosoh Bioscience, Griesheim, Germany) column (4.6 × 250 mm, 5 μm). The detection of eluted compounds was performed simultaneously with a flow cell diode array UV-vis detector and an ESI-q-TOF high-resolution hybrid mass spectrometer maXis 4G (Bruker Daltonics, Bremen, Germany) connected to the chromatograph. The collected OSSO solution was aliquoted, lyophilized and frozen at −70 °C until use.

### 2.2. Optical Measurements

Steady-state and time-resolved UV-Visible absorption measurements were performed using an Agilent 8453 spectrophotometer from Hewlett-Packard (La Jolla, CA, USA). All measurements were carried out in a 10 × 10 mm^2^ quartz cell. During the time-resolved measurements, optical absorption spectra were recorded at 1 s intervals. To study the kinetics of the process, current optical density at 6 fixed wavelengths (220 nm, 240 nm, 265 nm, 300 nm, 310 nm and 320 nm) was displayed. The required temperature in the cell was maintained with an Agilent 89090A (La Jolla, CA, USA) temperature control unit (available temperature range from −10 °C to 120 °C). During the experiment, the solution in the cell was constantly stirred with a magnetic stirrer. All solutions were bubbled with argon for 15 min prior to optical measurements.

### 2.3. Laser Flash Photolysis Measurements

The nanosecond laser flash photolysis (LFP) setup was described previously [32]. Briefly, sample excitation in a 10 × 8 mm^2^ quartz cell was performed using a Quanta-Ray LAB-130-10 Nd:YAG laser from SpectraPhysics (Mountain View, CA, USA) at 355 nm and the detection was performed using light from a DKSh-150 xenon short-arc lamp from Stella Ltd. (Moscow, Russia). All solutions were bubbled with argon for 15 min prior to and during irradiation.

### 2.4. NMR Measurements

1H NMR measurements were carried out at the Center of Collective Use «Mass spectrometric investigations» SB RAS with the use of a NMR spectrometer AVANCE III HD 700 MHz (Bruker BioSpin, Rheinstetten, Germany) equipped with a 16.44 Tesla Ascend cryomagnet, as described previously [13]. The solutions for NMR measurements were prepared in 20 mM deuterated phosphate buffer (pH 7.2) containing 2 × 10^−5^ M sodium 4,4-dimethyl-4-silapentane-1-sulfonic acid (DSS) as an internal standard.

## 3. Results

### 3.1. Absorption Spectra of OSSO and OSH

Dry OSSO (the structure is shown in Appendix A) was dissolved in PBS and the UV-Vis absorption spectrum was measured (Figure 1). The concentration of OSSO in solution and its purity was controlled using NMR spectroscopy by integration of the OSSO signal relatively to the DSS signal. The obtained spectrum had a maximum at 265 nm with ε_265_(OSSO) = 7780 M^−1^cm^−1^, which is in a good agreement with the spectrum published in [33]. Then, a two-fold amount of glutathione GSH was added to the OSSO solution and bubbled with argon for 30 min; then, the spectrum was measured. The OSH spectrum (Figure 1; ε_240_(OSH) = 10320 M^−1^cm^−1^) was obtained as a difference between the obtained spectrum and the spectrum of oxidized glutathione GSSG formed during the reaction of OSSO reduction by GSH.

### 3.2. OSSO Reduction by GSH

The reduction of OSSO to OSH upon interaction with GSH is a two-stage process proceeding through the formation of the OSSG (Appendix A) intermediate [4]:(5)$$\mathrm{OSSO}\;+\;\mathrm{GSH}\;\overset{k_{1}}{\to }\;\mathrm{OSSG}\;+\;\mathrm{OSH},$$
(6)$$\mathrm{OSSG}\;+\;\mathrm{GSH}\;\overset{k_{2}}{\to }\;\mathrm{GSSG}\;+\;\mathrm{OSH}.$$

The determination of the reaction rate constants k_1_ and k_2_ was performed with the use of kinetic optical measurements. To this purpose, a 3.7 × 10^−5^ M solution of OSSO in PBS (20 mM, pH of 7.4) was placed into the quartz cell of a spectrophotometer and bubbled for 15 min with argon; then, the excess GSH was added to the solution. During the experiment, the solution in the cell was constantly stirred with a magnetic stirrer and the temperature of 25 °C was maintained with a temperature control unit. The initial concentrations of GSH were 0.2 mM, 0.3 mM and 0.5 mM, i.e., in all cases, the concentration of GSH was significantly higher than that of OSSO. The monitoring of the optical absorption of the sample during the reaction was performed for six wavelengths (220 nm, 240 nm, 265 nm, 300 nm, 310 nm and 320 nm), where the difference in the absorption between OSSO and OSH was the highest (Figure 1).

Examples of the kinetic curves are shown in Figure 2. One can see two distinct processes, a fast reaction (5) completing within the first 50 s and a much slower reaction (6). Since the concentration of GSH was much higher than that of OSSO and OSSG, both reactions can be considered as pseudo-first order ones, k_1_′ = k_1_ × [GSH] and k_2_′ = k_2_ × [GSH]. The obtained kinetic curves were fitted to biexponential function; the obtained results are shown in Figure 2. The calculated values of the rate constants k_1_ and k_2_ are given in Table 1.

The same measurements were performed for two other temperatures, 15 °C and 5 °C. The Arrhenius plots for rate constants k_1_ and k_2_ are shown in Figure 2 and the obtained Arrhenius parameters are A_1_ = (2.2 ± 0.2) × 10^9^ M^−1^s^−1^, E_a1_ = 36 ± 4 kJ/mol, A_2_ = (3.8 ± 0.2) × 10^8^ M^−1^s^−1^ and E_a2_ = 41 ± 4 kJ/mol. Taking into account that the measurements were performed for a relatively narrow temperature range, from 5 °C to 25 °C, the obtained parameters should be considered as a good estimation rather than the exact data.

The measurements of the k_1_ and k_2_ dependences on pH were performed only for T = 25 °C. At a pH of 7.2, the obtained values were k_1_ = 520 M^−1^s^−1^ and k_2_ = 20.5 M^−1^s^−1^; the pH decrease, down to 6.8, resulted in the rate constant decrease to k_1_ = 240 M^−1^s^−1^ and k_2_ = 12.8 M^−1^s^−1^ (Table 1). Apparently, this effect should be attributed to the reduced deprotonation of GSH at low pH values. The pKa value of the SH group of glutathione was equal to 8.7 and the percentage of chemically active thiolate form GS^-^ decreased from 5% at a pH of 7.4 to 3.2% at a pH of 7.2 and to 1.3% at a pH of 6.8.

The optical absorption of the solution during the reaction was determined by the concentrations and electronic spectra of the reactants; at every wavelength and time point, the observed optical density of solution can be expressed as
OD = ε_(OSSO)_[OSSO] + ε_(GSH)_[GSH] + ε_(OSSG)_[OSSG] + ε_(OSH)_[OSH] + ε_(GSSG)_[GSSG],(7)
where ε is the absorption coefficient.

The absorption spectra of OSH, OSSO (Figure 1), GSH and GSSG are already known and the concentrations of compounds at every time point can be calculated using the values of the rate constants k_1_ and k_2_. Thus, the only unknown variable in Equation (7) is the absorption spectrum of the intermediate OSSG. This spectrum was obtained by subtraction of the calculated optical densities of OSH, GSH, OSH and GSSG from the observed optical densities of the solution OD for three time points, 50 s, 139 s and 189 s. For all three time points, similar spectra of OSSG have been obtained; the resulting spectrum in units of absorption coefficient is shown in Figure 1. Similarly to OSH, the spectrum of OSSG had an absorption maximum at 240 nm with ε_240_ = 7310 M^−1^cm^−1^.

### 3.3. OSH Oxidation by Hydrogen Peroxide

To obtain the reduced ovothiol OSH, we dissolved the OSSO extracted from the fish lens in PBS (pH of 7.4, 20 mM). The solution concentration measured by OSSO absorption at 265 nm (ε_265_ = 7780 M^−1^cm^−1^) was adjusted to 2.3 × 10^−5^ M and then 4.6 × 10^−5^ M GSH was added. The solution was stirred under argon bubbling for 30 min to achieve the complete transformation of OSSO into OSH via the reaction with GSH.

The kinetic measurements were performed in the same way as we did in the OSSO reduction experiments; the solution of OSH was placed into the cell of the spectrophotometer and then the excess of H_2_O_2_ was added. The optical absorption of the solution was monitored at the wavelengths 240 nm, 265 nm, 300 nm, 310 nm and 320 nm; typical kinetic curves are shown in Figure 3. The measurements were performed for three temperatures, 25 °C, 15 °C and 5 °C. The concentrations of added H_2_O_2_ were 3 mM, 6 mM and 9 mM.

It was found that the signal decay at 240 nm and 265 nm and signal growth at 300 nm, 310 nm and 320 nm were exponential and their rates linearly depended on the H_2_O_2_ concentration. At sufficiently high H_2_O_2_ concentrations, one can observe that the signal growth at 300 nm, 310 nm and 320 nm transforms into slow decay (Figure 3). The observed kinetics are in a good agreement with the reaction scheme of thiol reactions with H_2_O_2_ [34,35,36,37,38,39]:(8)$${\mathrm{OS}}^{-}\;+\;H_{2}O_{2}\;\overset{k_{3}}{\to }\;\mathrm{OSOH}+{\mathrm{HO}}^{-},$$
(9)$${\mathrm{OS}}^{-}\;+\;\mathrm{OSOH}\;\to \;\mathrm{OSSO}\;+\;{\mathrm{HO}}^{-},$$
(10)$$\mathrm{OSOH}\;+\;\mathrm{OSOH}\;\to \;\mathrm{OS}(O)\mathrm{SO}\;H_{2}O,$$
(11)$$\mathrm{OSOH}\;+\;H_{2}O_{2}\;\to \;{\mathrm{OSO}}_{2}H\;+\;H_{2}O.$$

According to this scheme, the initial spectral variations correspond to the reaction (8) of OS^−^ with H_2_O_2_, followed by a significantly slower decay of sulfenic acid OSOH (Appendix A). The initial parts of the kinetic curves were treated as exponential functions, with k_3_′ = k_3_ × [H_2_O_2_]; the values of the second order rate constant k_3_ were calculated from the slope of the k_3_′ dependence on the H_2_O_2_ concentration (Appendix A). The obtained values of k_3_ for 25 °C, 15 °C and 5 °C are collected in Table 1. The Arrhenius plot (Figure 3) for k_3_ yielded A_3_ = (7.2 ± 0.7) × 10^8^ M^−1^s^−1^ and E_a3_ = 47 ± 4 kJ/mol.

The slow decay of the OSOH signal monitored at 265 nm, 300 nm, 310 nm and 320 nm did not depend on the H_2_O_2_ concentration and, therefore, cannot be attributed to the reaction (11). The values of the decay rate constant k_4_ obtained from the exponential fit of the far parts of the kinetics are given in Table 1. One can see that the signal decay continues even when all OSH is already consumed in the reaction with hydrogen peroxide, so the involvement of the reaction (9) is also rather unlikely. Most likely, the signal decay corresponds to the bimolecular reaction (10) or other unknown reactions of sulfenic acid OSOH. We should notice that, under our experimental conditions, only the initial parts of kinetics of OSOH decay were recorded (especially for samples with low H_2_O_2_ concentrations) and the obtained k_4_ values are rather imprecise.

The pH decrease down to 6.8 did not result in noticeable changes in the rate of reaction (8). That is not surprising, since OSH remains in the thiolate form in a broad pH range.

At the time point of 156 s, practically all OSH had already been converted into OSOH, while the decay of OSOH had only started. Thus, at this time point, the only compounds present in the solution were hydrogen peroxide and sulfenic acid and the difference between the observed absorption spectrum at t = 156 s and the spectrum of H_2_O_2_ yielded the absorption spectrum of OSOH. The obtained spectrum is shown in Figure 1.

### 3.4. Triplet Quenching by OSH

To study the reaction of the triplet-state quenching by OSH, we took kynurenic acid (KNA, Figure 1) as a model photosensitizer due to the very convenient properties of this compound; KNA readily dissolves in water, absorbs in the UV-B region and, under irradiation, it efficiently (quantum yield of 0.82) forms a triplet state with the strong absorption band at 600 nm [40]. LFP measurements were performed for 2.6 × 10^−4^ M KNA solution in PBS (pH of 7.2) in the absence and presence of OSH in solution. The KNA concentration of 2.6 × 10^−4^ M provided optimal absorption (OD = 0.83) in the photolytic cell at 355 nm, the wavelength of the laser irradiation.

Solutions of KNA with OSH were prepared by addition of OSSO and excess of GSH into the KNA solution. The presence of the rest of GSH in solution did not influence the kinetics of ^T^KNA quenching, since GSH at a neutral pH is a rather bad triplet quencher [30] and does not absorb in the UV-A region. The final concentrations of OSH were 1 × 10^−4^ M, 2 × 10^−4^ M and 3 × 10^−4^ M.

The kinetics of ^T^KNA decay was detected at 600 nm. For every OSH concentration, the measurements were performed 4–5 times with laser energy varying from 1 to 5.9 mJ/pulse in order to separate the contributions from the first and second order reactions. Figure 4 demonstrates the kinetic curves obtained with different OSH concentrations. Without triplet quenchers, ^T^KNA decays almost purely by the second order reaction of triplet annihilation [30,40]. In the presence of OSH, the triplet decay significantly accelerates and becomes exponential. The Stern–Volmer plot of the dependence of the observed pseudo-first order rate constant of triplet decay on the OSH concentration (inset in Figure 4) yields the second order rate constant of triplet KNA quenching by OSH, k_q_ = (1.9 ± 0.1) × 10^9^ M^−1^s^−1^.

## 4. Discussion

The results of the present work demonstrate that, in biological systems, the combined actions of OSH and GSH provide very reliable antioxidant protection. At neutral conditions (pH of 7.4) and T = 25 °C, the rate constant of the OSH reaction with hydrogen peroxide (3.7 M^−1^s^−1^) is four-fold higher than that for GSH (0.87 M^−1^s^−1^, pH of 7.4 [41]). It should be noted that the published data on the rate constant of the GSH reaction with H_2_O_2_ range from 0.43 M^−1^s^−1^ (pH of 7.2) [42] to 1.6 M^−1^s^−1^ (pH not shown) [6]. Most likely, these differences correspond to the high sensitivity of the reaction to the solution pH. The rate constant data can be combined with the real levels of antioxidants in living nature. For example, the concentration of OSH in the lens of *Sander lucioperca* varies between 1.5 mM and 3 mM and the GSH level is approximately 0.5 mM [26,27]. Thus, one can estimate that the reduction of H_2_O_2_ by GSH alone would take more than half an hour, while the presence of OSH reduces this time down to 2–3 min. Oxidized ovothiol OSSO is then reduced by GSH back to OSH. Taking into account the values of the rate constants k_1_ and k_2_ measured in this work, the complete restoration of OSH occurs within several minutes. Previously, it has been shown that OSH and its analogs can effectively scavenge other ROS, such as superoxide [5] and hydroxy radical [9].

The rate-constant values obtained in the present work can be compared with the previously published data. The reaction between OSH and hydrogen peroxide was studied in works [6,42]. The reported values k_3_ = 2 M^−1^s^−1^ [42] and 3.18 M^−1^s^−1^ [6] are in a fair agreement with our results. The rate constants k_1_ of the GSH reaction with OSSO and k_2_ of the GSH reaction with OSSG are reported in this work for the first time, as well as the absorption spectra of reaction intermediates OSSG and OSOH.

The important finding of the present work is that OSH is an excellent triplet-state quencher; the reaction between triplet KNA and OSH proceeds with the rate constant k_q_ = 1.9 × 10^9^ M^−1^s^−1^. In mammalian eye lens, the quenching of triplet states is provided by high concentrations of ascorbate [29,31,43,44], which reacts with triplet KNA with an almost diffusion-controlled rate k_q_ = 1.4 × 10^9^ M^−1^s^−1^ [30]. For instance, the level of ascorbate in the human lens is approximately 0.5–1.0 mM [29,31,44]. The concentration of OSH in the fish lens is even higher than the level of ascorbate in the human lens and the rate constant of the triplet quenching by OSH is higher than that by ascorbate. Besides, photo-induced oxidation of ascorbate inside the lens results in the formation of dehydroascorbate, which eventually leads to the accumulation of advanced glycation end products [45]. The decay of OS^•^ radicals most likely proceeds via the radical combination with the formation of harmless OSSO. Therefore, OSH in the fish lens provides at least as good a protection against photo-generated triplet states as ascorbate in the human lens. Previously [46], it has been shown that the rate constants of the triplet-state quenching by other thiols (cysteine and GSH) increases with the pH increase. This indicates that deprotonated thiols readily donate electron to triplet molecules and a high rate of the triplet-state quenching by OSH at neutral pH values should be attributed to the very low pKa value of the SH group of this compound.

We have recently shown [26] that the OSH level in lenses of freshwater fish during the winter undergoes significant seasonal variations; the OSH concentration in late autumn is 2–3 times higher than that in early spring. This finding was attributed to the decreased level of dissolved oxygen in ice-covered water bodies, decreased feeding activity of fish and, correspondingly, deceleration of metabolic processes in fish, including the OSH accumulation. However, an alternative explanation of this effect might be almost complete darkness in water under ice. Without solar radiation, the generation of triplet states, radicals and ROS decreases, causing the decrease in OSH production as a response. In recent works [21,47], it has been shown that the expression of the gene encoding the key ovothiol biosynthetic enzyme, ovoA, in diatoms depends on the light conditions. The enhanced expression of ovoA under light irradiation can be explained as the need of its presence for both reduction of light-induced ROS and quenching of reactive triplet states.

Unique antioxidant properties of OSH cause increased interest for its use in medicine. Pilot studies on this subject were performed with the use of cell cultures [12,33,48]. In particular, it has been demonstrated [33] that OSH reduces proliferation of human liver carcinoma cells and does not affect normal human embryonic lung cells. An especially promising direction of the use OSH in medicine is the treatment of diseases associated with oxidative stress and inflammation, such as diabetes or cardiovascular diseases. In a recent paper [12], it was shown that oxidized ovothiol OSSO is readily taken up by human endothelial cells, transforms into its reduced form (OSH) inside the cells and then acts as an antioxidant scavenging reactive oxygen and nitrogen species. These results indicate the therapeutic potential of OSH in the treatment of a broad range of diseases from tumors to cardiovascular diseases. We believe that the results obtained in the present work are useful both for a better understanding of redox reactions occurring in living nature and for a development of new medicine for treatment of human diseases.

## 5. Conclusions

The results obtained in this work demonstrate the unique functionality of ovothiol A and emphasize the importance of further studies on this metabolite. By entering cells as an external antioxidant and photoprotector, it can effectively integrate into the intracellular antioxidant defense system based on glutathione. Then, acting together with glutathione, ovothiol A is able to actively scavenge ROS and free radicals, as well as quenching dangerous triplet states, thus inhibiting oxidative stress and photoinduced damage. Taking into account the deleterious effect of oxidative stress for cells and tissues, OSH has great therapeutic and pharmacological potential.

## Data Availability

The data obtained in this study, including UV-Vis spectra and kinetic data, are available on request from the corresponding author. The data are not publicly available due to the large number of spectra and kinetic curves obtained.

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
