# Peer review of "Kinetic Studies of Antioxidant Properties of Ovothiol A"

_antioxidants, 2021, doi:10.3390/antiox10091470_

Round 1

Reviewer 1 Report

This paper reports a kinetic study on the redox properties of ovothiol A (oxidation by H2O2 to its oxidized form OSSO, reduction of OSSO by glutathione and quenching of the triplet state of the model photosensitizer kynurenic acid.

The study is well designed and performed. It reports interesting new results, considering that ovothiol is a natural antioxidant whose redox properties have been less studied than other reducing agents.  Overall, the author conclusions are nicely supported by the experimental data.   I believe that this paper is suitable for publication in antioxidant.

I have just few minor criticisms.

The drawing of some relevant compounds subject of this paper (OSSO, kynurenic acid) should be added in the introduction chapter.

I note that the Arrhenius parameters for the different reactions analysed have been calculated by using only a limited number of data points (see the K1 and K2 values at 25°C in Table 1, Figures 2 and 3). Possibly, one or two data points could be added. Otherwise, the authors should clearly state in the ‘Results’ chapter that the Arrhenius parameters are estimated and an error on the reported values should be added.

For readers less expert in the study of photosensitizing agents, the authors should briefly explain also in the introduction the ‘rationale’ at the basis of their study, at the basis of their detailed discussion at page 8 and 9.

Author Response

Comment: The drawing of some relevant compounds subject of this paper (OSSO, kynurenic acid) should be added in the introduction chapter.

Reply: We added the structure of kynurenic acid to Scheme 1, and the structures of GSH, OSSO, OSSG, and OSOH are now given in Scheme S1 (Supplementary Information).

Comment: I note that the Arrhenius parameters for the different reactions analysed have been calculated by using only a limited number of data points (see the K1 and K2 values at 25°C in Table 1, Figures 2 and 3). Possibly, one or two data points could be added. Otherwise, the authors should clearly state in the ‘Results’ chapter that the Arrhenius parameters are estimated and an error on the reported values should be added.

Reply: Limited number of temperature data points is determined by limited amount of available OSH, because this compound was extracted and chromatographically separated from the pike-perch lenses. In the revised version of the manuscript, the error values for the Arrhenius parameters are now given. At lines 164-166, the following statement is present: “Taking into account that the measurements were performed for relatively narrow temperature range from 5 °C to 25 °C, the obtained parameters should be considered as a good estimation rather than the exact data.”

Comment: For readers less expert in the study of photosensitizing agents, the authors should briefly explain also in the introduction the ‘rationale’ at the basis of their study, at the basis of their detailed discussion at page 8 and 9.

Reply: The following sentence is now added at the last paragraph of the introduction: “For better understanding of chemical and biochemical processes occurring in the fish lens under oxidative stress or UV irradiation, one needs to know the mechanisms and the rate constants of both thermal reactions of OSH oxidation and OSSO reduction and of photochemical reactions involving OSH”.

Reviewer 2 Report

The present paper describes an interesting analysis of the antioxidant properties of ovothiol A. The results of the kinetic analyses are interesting validating its publication in Antioxidants. However, before that, a few points need to be clarified:

  1. The abbreviations are confusing. Please define exactly the well-known abbreviations and acronyms too. Furthermore, the structure of OSSO, OSOH, OSSG must be defined and insert.
  2. The quality of the manuscript is decreased by the following errors due to negligence:

- Scheme 1 is not mentioned in the text.

- The pH of the PBS is not written in 2.1. Materials but it is continuously mentioned in 3. Results.

- The interpretation of OSOH.3.2. is missing (Fig. 1.) In general, the caption of figure 1 is confusing.

- The rate constant k2 is assigned to two reactions (eqs. 5 and 6) while k1 is not defined.

- The rate constant k4 is also not defined, however, its data are inserted into Table 1.

- The numbering in Section 3 is incorrect, Section 3.2 is missing.

- etc

  1. It is not clear to me, that the spectra depicted in Fig.1. are measured or constructed? Furthermore, the spectra of OSSO, OSH, and OSSG support the understanding of how the rate constants of OSSO reduction to OSH have been determinate. But why is the spectrum of OSOH depicted in this figure?
  2. The data of Table 1. can be separated into two tables, because the OSSO reduction is analyzed in Section 3.1 while the OSH oxidation is in Section 3.3.
  3. The last paragraph of the discussion maybe fitted better to the introduction.

Author Response

Comment: The abbreviations are confusing. Please define exactly the well-known abbreviations and acronyms too. Furthermore, the structure of OSSO, OSOH, OSSG must be defined and insert.

Reply: The structures of all compounds mentioned in the manuscript are now given in either Scheme 1 or Scheme S1.

Comment: Scheme 1 is not mentioned in the text.

Reply: It is mentioned at the very first line of the Introduction.

Comment: The pH of the PBS is not written in 2.1. Materials but it is continuously mentioned in 3. Results.

Reply: In this work, we used several buffers with different pH’s (6.8, 7.2, and 7.4). For this reason, it seems more convenient to indicate pH values for specific measurements in the Results section.

Comment: The interpretation of OSOH.3.2. is missing (Fig. 1.) In general, the caption of figure 1 is confusing.

- The rate constant k2 is assigned to two reactions (eqs. 5 and 6) while k1 is not defined.

- The numbering in Section 3 is incorrect, Section 3.2 is missing.

Reply: Thanks for indicating these mistakes, the caption to Fig. 1, the section numbering, and the eq.5 are now corrected.

Comment: The rate constant k4 is also not defined, however, its data are inserted into Table 1.

Reply: The definition of k4 is given at the line 217 of the manuscript.

Comment: It is not clear to me, that the spectra depicted in Fig.1. are measured or constructed?

Reply: The spectrum of OSSO was measured directly, while the spectra of OSH, OSSG, and OSOH were constructed according to procedures described at lines 134-136, 183-187, and 227-231.

Comment: Furthermore, the spectra of OSSO, OSH, and OSSG support the understanding of how the rate constants of OSSO reduction to OSH have been determinate. But why is the spectrum of OSOH depicted in this figure?

Reply: OSOH is an unstable compound, and its spectrum has not been published before. We decided that including this spectrum into present paper might be useful for other researchers working in this field.

Reviewer 3 Report

This work by Osik et al describes the kinetics of the reaction of ovothiol A (OSH), one of the most potent natural antioxidant, with H2O2, and the reduction of the disulphide (OSSO) by glutathione.  The compound has been isolated from fish lens according to a protocol previously developed by the same authors in the form of the disulphide and reduced by addition of glutathione. Moreover,  the reported ability of OSH to act as efficient quencher of triplet state of kynurenic acid further adds to the protective action that this compound may exert in the tissues like fish lenses in which it occurs at millimolar levels. 

Though the peculiarity of OSH in terms of pKa and oxidizability has been widely described in previous works including  its reduction by glutathione I feel that this study is of interest because further supports these early observations on a quantitative ground by determination of the rate constants. Presentation of results is very clear and both introduction and discussion well summarize previous results and correctly frame those of the present study in the literature . 

I have only minor suggestions:

For the sake of clarity show OSH at the ionization state prevailing at physiological pHs in scheme 1

Eq. 1 use equilibrium arrows not double sided arrow used to indicate mesomers

Discussion:  please consider reaction of OSH with ROS other than H2O2 and discuss data from literature if available

Figure 3 H2O2 used is at enormous excess with respect to OSH but in the same order of magnitude;  why not to use a wider range of variations that is orders of magnitude to see expectedly larger differences in the kinetics?

Author Response

Comment: For the sake of clarity show OSH at the ionization state prevailing at physiological pHs in scheme 1

Reply: Throughout the whole manuscript, we used abbreviation OSH to indicate reduced ovothiol in general, and the short name OS- is used only in cases when the deprotonated state of the compound is considered. In our opinion, it is more logical to present in Scheme 1 the structure corresponding to OSH molecule rather than to OS- anion.

Comment: Eq. 1 use equilibrium arrows not double sided arrow used to indicate mesomers

Reply: The type of arrows is now changed.

Comment: Discussion:  please consider reaction of OSH with ROS other than H2O2 and discuss data from literature if available

Reply: We added into discussion (lines 283-285) the following sentence: “Earlier, it has been shown that OSH and its analogs can effectively scavenge other ROS such as superoxide [5] and hydroxy radical [9]”.

Comment: Figure 3 H2O2 used is at enormous excess with respect to OSH but in the same order of magnitude;  why not to use a wider range of variations that is orders of magnitude to see expectedly larger differences in the kinetics?

Reply: The range of H2O2 concentrations used in this study is determined by experimental conditions: if [H2O2] is too high, the absorption of this compound impedes the measurements at short wavelengths. If it is too low, the rate of signal increase (Figure 3) becomes comparable or even lower than the signal decay, which complicates the data analysis. Therefore, we selected the concentration range from 3 to 9 mM, which is sufficient for the reliable rate constant measurements.